# Seasonal Impact on Free Flap Surgery in Terms of Flap Loss and Wound Healing Disorders: A Retrospective Cohort Study of 158 Free Flaps

**DOI:** 10.3390/healthcare11030403

**Published:** 2023-01-31

**Authors:** Marie-Luise Klietz, Chiara Mewe, Philipp Wiebringhaus, Tobias Hirsch, Matthias Michael Aitzetmüller, Maximilian Kückelhaus

**Affiliations:** 1Department for Plastic and Reconstructive Surgery, Institut for Muskuloskeletal Medicine, Westfälische Wilhelms-University, 48149 Münster, Germany; 2Department for Plastic and Reconstructive Surgery, Fachklinik Hornheide, 48157 Münster, Germany; 3Division for Plastic Surgery, Department for Traumatology and Hand Surgery, University Hospital Münster, Albert-Schweitzer-Campus 1, 48149 Münster, Germany

**Keywords:** seasonal impact, free flap surgery, breast reconstruction

## Abstract

(1) Background: Postoperative flap loss and wound healing disorders are severe complications after microsurgical free flap surgery. Despite multiple clinical observations, a possible influence of season and external temperature on outcome are largely missing. (2) Methods: Retrospectively, data were collected from 151 patients receiving microsurgical free flaps from March 2018 to August 2019. Patients were divided into two cohorts. The winter group includes all patients who underwent surgery from October 2018 to March 2019 and the summer group al those who underwent surgery from April 2018 to September 2018. Data included demographic information, pre-existing conditions, flap characteristics, and postoperative complications like flap losses and wound healing problems. External temperatures during the first 14 postoperative days were documented and the predictor of flap loss and wound healing disorders was detected. (3) Results: In the winter group (October-March; Ø 7.24 °C) 72 patients (46 female, 24 males; Ø 57.0 years) and in the summer group (April-September; Ø 18.79 °C) 81 patients (48 female, 33 males; Ø 56.0 years) received free flap surgery. There were no significant differences in demography (age: *p* = 0.593; gender: *p* = 0.419; BMI: *p* = 0.141). We found a significant increase in flap loss during summer (χ^2^(1) = 6.626; *p* = 0.010; V = 0.209) strengthened by logistic regression analysis (*p* = 0.037; Exp(B) = 9.655). Additionally higher average temperatures 14 days postoperatively represents another main driver (*p* = 0.023, Exp(B) = 1.161) for postoperative flap loss. (4) Conclusions: The data confirm a significantly higher postoperative flap loss in the summer group. This information may potentially contribute to optimization of perioperative management and planning of elective and semi-elective surgeries.

## 1. Introduction

Since their popularization in the 1970s, free flap reconstruction is still gaining in importance in Plastic and Reconstructive Surgery. They represent salvage procedures for extremities or reconstruction methods of choice after cancer.

While free flaps after traumas are hardly plannable, reconstruction after breast cancer represents a semi-elective procedure. Therefore, potential risk factors that may increase the prevalence of complications should be minimized prior surgery. For example, smoking should be ceased, and Tamoxifen as a potential pro-thrombotic agent should be paused for the surgery due to enhanced flap loss rate [1].

Possible specific complications of defect coverage by free flap surgery include flap loss and postoperative wound healing disorders. There are numerous data on complication rates in the literature, but no discussion of the seasonal relationship. However, seasonal impact on complication rates has been described in other types of surgery such as body contouring procedures [2,3] or spine surgeries [4]. Regarding to body contouring procedures, the season was even shown to be the strongest predictor of complications, potentially explained by seasonal changes in microbial skin flora [5]. Nevertheless, the influence of seasons has not yet been investigated in flap reconstructions—which sometimes represent a semi-selective operation and whose timing can be planned and influenced in relation to breast reconstruction for breast carcinoma. Besides the possibility of enhanced wound healing disorders caused by changes in skin flora, higher temperature could potentially lead to dehydration that might result in enhanced thrombotic events—a phenomena that already has been described for cerebral insults [6,7].

Therefore, the aim of this work is to investigate the seasonal effects on the occurrence of complications after free flap surgery.

In this regard, all patients who underwent free flap procedure in our department between March 2018 and October 2019 were retrospectively analyzed.

## 2. Materials and Methods

### 2.1. Ethical Approval

Written approval from the Ethics Committee is available (2020832-f-S).

### 2.2. Study Design

In this retrospective monocentric cohort study, 151 patients who received 158 free flap reconstruction at our department between March 2018 and October 2019 were included. Patients who were transferred to peripheral hospitals and thus did not undergo postoperative follow-up were excluded.

Patient data were collected retrospectively based on the patient files in the hospital information system (HIS) “iMedOne”, the Department of Plastic, Reconstructive and Aesthetic Surgery of the Hornheide Clinic. The study was approved by the local Ethical Committee (2020-832-f-S).

### 2.3. Demographic Data

Demographic data included the following patient related factors: age, gender, BMI, diabetes mellitus, hypertension, arteriosclerosis, anemia, postoperative antibiotic therapy, smoking, and postoperative C-reactive protein value in mg/L.

The indications for defect coverage were divided into the following groups: soft tissue defect after tumor excision, soft tissue defect due to infection, and soft tissue defect due to trauma. The location of the defect was differentiated into lower extremity, upper extremity, and mammary/abdominal. The following flaps were performed: Anterolateral Thigh Perforator (ALT) flap, Latissimus dorsi muscle flap, Deep Inferior Epigastric Artery Perforator (DIEP) flap, Profunda Artery Perforator (PAP) flap, Transverse Rectus Abdominis Musculocutaneus (TRAM) flap, Superficial Circumflexa Iliaca (SCIP) flap, and Free Gracilis Muscle Flap.

A distinction was made between the connecting vessel and the type of anastomosis (end-to-end vs. end-to-side).

### 2.4. Outcome Parameters

The season as well as the temperature were used for cohort selection. All patients who underwent surgery from 1 November 2018 to 30 April 2019 were assigned to the winter group. All patients who underwent surgery between 1 May 2019 and 31 October 2019 were assigned to the summer group.

In order to be able to determine the seasonal impact on complications in terms of wound healing disorders and flap loss, average outdoor temperatures were recorded both on the day of surgery and 14 days postoperatively. The source is based on data from international weather models, satellite imagery, and radar information [8]. Further primary outcome parameters represent flap and donor site-related complications, partial and complete flap loss, and prevalence of thrombotic events such as systematic complications.

### 2.5. Statistical Analysis

The collected parameters were entered into a Microsoft Excel 2019, version 16.0 spreadsheet (Microsoft Excel, Redmond, WA, USA). Numerical coding was performed. All collected data were entered into IBM SPSS Statistics (IBM SPSS Statistics v27, Armonk, NY, USA) using the Excel spreadsheet and statistically analyzed. In order to record the differences of the epidemiological data regarding the winter group and the summer group, a T-test for independent samples was performed. To analyze the relationship between the revision rate and a resulting wound healing disorder, as well as for the relationship between revision rate and any resulting flap loss, a chi-square test was performed. For group comparison, an independent sample test was performed. For risk calculation for flap loss, a logistic regression analysis was performed for the independent parameters. Significance was calculated at the 5% level (*p* < 0.05).

## 3. Results

A total of 158 free flaps were performed in 151 patients—epidemiological data are given in Table 1.

### 3.1. Postoperative Data

The difference of age, gender and BMI in winter group and summer group was not significant between groups (age: t(156) = 0.593, *p* = 0.593 > 0.05/gender: (t(156) = −0.811, *p* = 0.419 > 0.05/bmi: (t(156) = −1.478, *p* = 0.141 > 0.05)). Variance homogeeity for the variables of age, sex and BMI is shown in Table 2.

### 3.2. Revision Rates and Wound Healing Disorders

There was no statistically significant association between revision rate and wound healing disorder (χ^2^(5) = 5.74, *p* = 0.33, V = 0.197).

### 3.3. Wound Healing Disorders—Comparison of Winter Group and Summer Group

Overall, 35% (*n* = 29) of the 83 patients treated in summer and 49% (*n* = 37) of the 75 patients treated in winter (M = 0.49; SD = 0.503) suffered from a wound healing disorder at the flap (M = 0.35; SD = 0.480). There was no variance homogenity for wound healing disorder (*p* = 0.007 < 0.05) and a marginally statistically significant increase in wound healing disorders in the winter group (t(153) = 1.835, *p* = 0.068 > 0.05).

### 3.4. Revision Rates and Flap Loss

There was a statistically significant relationship between the frequency of revisions and flap loss (χ^2^(5) = 46.74, *p* = 0.000, V = 0.5629). Thus, the revision rate was shown to have a strong effect on flap loss.

### 3.5. Flap Losses—Comparison Winter Group and Summer Group

The independent samples test showed that group-wise 12% (*n* = 10) of the 83 patients treated in summer suffered flap loss (M = 0.12; SD = 0.328) and 1% (*n* = 1) of the 75 patients treated in winter (M = 0.01;SD = 0.115), indicating a statistically significant increase in flap loss in the summer group (t(104) = −2.795; *p* = 0.006 < 0.05).

Overall, the season of the year was found to be the strongest predictor for the development of postoperative flap loss in summer. Table 3 shows logistic regression analysis and siginificance of the model with the variable “season” as the strongest predictor for flap loss.

### 3.6. Temperatures of the 14 Postoperative Days and Flap Loss

Logistic regression analysis showed the significance of the model (chi-square(1) = 6.636, *p* = 0.010 < 0.05, *n* = 158)—it is shown in Table 4. 

It is shown that temperatures 14 days postoperatively were significant for the development of postoperative flap loss with *p* = 0.023 (Exp(B) = 1.161).

Hereby, flap loss was more likely to occur at elevated temperatures. Overall, the temperatures of the 14 postoperative days postoperatively could be determined as the strongest predictor for the development of postoperative flap loss in summer.

## 4. Discussion

As we already know form recent data, patient dependent factors such as smoking, BMI, or co-morbidities can significantly influence outcome of specific surgeries. This has been proven for body contouring procedures [2], spine surgeries [4,8,9], as well as for surgical site infections in general [10,11].

Regarding body contouring procedures, a recent analysis has identified the season as the highest predictor for surgical site infections, with an odds ratio of 2.693—being higher than age, smoking history, or BMI [2]. Similar to body contouring procedures, some free flap reconstructions, such as breast reconstruction, with autologous tissue sometimes represent semi-elective surgeries. Based on this pre-work, our analysis was performed to enhance safety and decrease adverse side effects of these patients. For comparison, a winter group and a summer group were formed based on the average monthly temperatures. Due to the fact that in our hospital, no air-cooling system has been installed in patient rooms, we estimate the outdoor temperature to be not equal but highly correlated to the local room temperature.

There was no statistical difference in group comparison regarding to age, gender, BMI, and key co-morbidities such as diabetes mellitus, smoking, anaemia, hypertension, or arteriosclerosis between the summer and winter group, and no difference was found in the etiology of the soft tissue defect, the free flaps that have been used, or the connecting vessels. Therefore, we considered our groups to be comparable.

Interestingly, contrary to what we have expected and what has been described in previous studies, there was no significant difference when comparing surgical site infections or wound healing disorders at the donor site. Only BMI and female gender showed significantly enhanced prevalence of SSIs and wound healing disorders.

When evaluating flap loss, a flap loss rate of 7.3% was found, being comparable to recent meta-analyses [12]. We found significantly enhanced flap loss rate in the summer season (*p* = 0.037), even being enhanced when only evaluating the first 14 days post-surgery (*p* = 0.027), indicating that temperature influences the outcome of free flap reconstructions.

This phenomenon may be explained by enhanced thrombotic events during summer months. Although not being specifically described for microsurgery, enhanced rates of cerebral thrombotic events during summer seasons are described in studies from India and Iran [6,7]. A study group from Pakistan describes a higher prevalence of thrombotic events that might be “influenced by seasonal changes in atmospheric temperatures and humidity, especially in patients with underlying prothrombotic risk factors” [13]. An underlying effect could be the heat-regulated enhanced circulation of pro-inflammatory cytokines such as Interleikin-6 that could lead to injures of blood vessel’s endothelial cells and thereby serve as further pro-thrombotic factor [13,14]. While the climate is not transferrable to Germany, the etiology may be similar. In both papers, dehydration is noticed as one risk factor for thrombosis. To dig deeper and to evaluate this pathomechanism in our cohort, we additionally analyzed the prevalence of arterial and venous thromboses. On average, arterial thrombosis at the anastomosis occurred in 8.2%, and venous thrombosis occurred at the anastomosis in 8.2%. The arterial thrombosis rate was 2.8 times higher in the summer group while venous thromboses showed a 2.6-fold increased rate in the summer group, supporting the hypothesis that a thrombotic event might worsen outcome in free flap surgeries during summer months. At this point, a comparison with continued anticoagulation as well with increased hydration during, prior and post-surgical intervention may be interesting.

While other studies show enhanced wound healing disorders driven by temperature and caused by changes in skin flora, this was only partially verified in our study. SSIs and wound healing disorders were only enhanced in the reconstructed region, but not in the donor region. Therefore, a change of skin flora and thereby enhanced infection rates may not fully explain this phenomenon. Nevertheless, bacterial contamination could trigger the immune system and cause thrombosis [15]. While changes in skin flora and thereby enhanced SSIs, wound healing disorders in the areas of reconstruction, and thrombosis may support flap loss, contamination as the only driver seems unlikely.

It must be mentioned that our study represents a retrospective data analysis and therefore, due to lack of data, the influence of adequate hydration or adaption of postoperative anti-thrombotic medication cannot be analyzed. This represents a limitation of our study.

Additionally, our study is a monocenter study, carried out in the North-West of Germany. Conclusions drawn from our study can only be transferred to centers with a similar climate.

Based on these results, many questions remain. While the exact pathomechanism is unclear, the effect of intensified anti-thrombotic medication or changes in hydration should further be investigated. Although meta-analysis shows only little effect by antithrombotic drugs such as low molecule weight heparin, Heparin, or ASS in free flap surgery [16], no data are available for extended antithrombotic therapy during summer months. Furthermore, there is no accepted and evidence-based standard for the management of anticoagulation in free flap surgery.

Additionally, there can remain a location-based bias such as specific air condition systems. Therefore, we would like to encourage other high-volume centers to assess the season being a risk factor for flap loss.

## 5. Conclusions

Although the exact pathomechanism remains unclear, this study provides evidence for the hot season being a risk factor for enhanced flap loss due to increase in venous and arterial thrombotic events. Larger multicentric prospective studies are necessary for confirmation and more insight into the involved pathomechanisms.

## Figures and Tables

**Table 1 healthcare-11-00403-t001:** Epidemiological data, * no statistically significances between winter and summer groups.

	Overall	Winter Group	Summer Group
**Age ***	Ø 56.41 years (±SD 5.39)	Ø 56,93 years (±SD 14.86)	Ø 55.96 years (±SD 15.93)
**Gender ***	62.3% female (*n* = 94) 37.7% male (*n* = 57)	65.7% female (*n* = 46) 34.3% male (*n* = 24)	59.3% female (*n* = 48) 40.7% male (*n* = 33)
**Co-Morbidities**			
Diabetes Mellitus *	11.3% (*n* = 17)	12.9% (*n* = 9)	9.9% (*n* = 8)
Hypertension *	33.8% (*n* = 51)	37.1% (*n* = 26)	30.9% (*n* = 25)
Arteriosclerosis *	15.2% (*n* = 23)	15.7% (*n* = 11)	14.8% (*n* = 12)
Anemia (WHO-criteria) *	61.6% (*n* = 93)	68.6% (*n* = 48)	55.6% (*n* = 45)
Nicotine Abuse *			
Non-Smoker	82.1% (*n* = 124)	84.3% (*n* = 59)	80.2% (*n* = 65)
<10 cigarettes per day	6.6% (*n* = 10)	4.3% (*n* = 3)	8.6% (*n* = 7)
10–20 cigarettes per day	7.3% (*n* = 11)	8.6% (*n* = 6)	6.2% (*n* = 5)
>20 cigarettes per day	4.0% (*n* = 6)	2.9% (*n* = 2)	4.9% (*n* = 4)
Body Mass Index *			
<20 kg/m2 (underweight)	0% (*n* = 0)	0% (*n* = 0)	0% (*n* = 0)
20–24 kg/m (normal weight)	41.1% (*n* = 62)	50.0% (*n* = 35)	33.3% (*n* = 27)
25–30 kg/m2 (overweight)	33.1% (*n* = 50)	27.1% (*n* = 19)	38.3% (*n* = 31)
>30 kg/m2 (obese)	25.8% (*n* = 39)	22.9% (*n* = 16)	28.4% (*n* = 23)
**C-reactive protein value in mg/L (postoperative) ***	55.5 (±SD 50.20)	73.91 (±SD 73.75)	103.48 (±SD 96.89)
**postoperative antibiotic therapy ***	36.4% (*n* = 55)	37.1% (*n* = 26)	35.8% (*n* = 29)
**number of postoperative hospitalized days ***	12.74 (±SD 9.41)	12.40 (±SD 7.67)	13.04 (±SD 10.75)
**indications for defect coverage ***			
trauma	33.5% (*n* = 53)	34.3% (*n* = 24)	35.8% (*n* = 29)
tumor	57.6% (*n* = 89)	58.6% (*n* = 41)	59.3% (*n* = 48)
infection	7.0% (*n* = 8)	7.1% (*n* = 5)	3.7% (*n* = 3)
**location of the defect ***			
lower extremity	49.4% (*n* = 74)	45.7% (*n* = 32)	51.9% (*n* = 42)
upper extremity	7.0% (*n* = 10)	8.6% (*n* = 6)	4.9% (*n* = 4)
mamma/abdomen	43.7% (*n* = 67)	45.7% (*n* = 32)	43.2% (*n* = 35)
**flap design ***			
ALT	32.3% (*n* = 51)	35.7% (*n* = 25)	32.1% (*n* = 26)
Latissimus-Dorsi	8.2% (*n* = 13)	7.1% (*n* = 5)	9.9% (*n* = 8)
DIEP	20.3% (*n* = 32)	20.0% (*n* = 14)	22.2% (*n* = 18)
PAP	20.9% (*n* = 33)	22.9% (*n* = 16)	21.0% (*n* = 17)
TRAM	1.3% (*n* = 2)	2.9% (*n* = 2)	0.0% (*n* = 0)
SCIP	11.4% (*n* = 18)	11.4% (*n* = 8)	12.3% (*n* = 10)
Gracilis	1.3% (*n* = 2)	0.0% (*n* = 0)	2.5% (*n* = 2)

**Table 2 healthcare-11-00403-t002:** Variance homogenity was present for the variables of age (*p* = 0.677 > 0.05), sex (*p* = 0.111 > 0.05) and BMI (*p* = 0.107 > 0.05).

	Overall	Winter Group	Summer Group
**average temperature on day of** **surgery**		9.09 °C	19.02 °C
vascular complications	8.2% (*n* = 13)	4.0% (*n* = 3)	12.0% (*n* = 10)
arterial thrombosis	8.2% (*n* = 13)	5.3% (*n* = 4)	10.8% (*n* = 9)
venous thrombosis	0% (*n* = 0)	0% (*n* = 0)	0% (*n* = 0)
postoperative flap loss	7.0% (*n* = 11)	1.3% (*n* = 1)	12.0% (*n* = 10)
wound healing disorders	43.1% (*n* = 65)	49.3% (*n* = 37)	34.9% (*n* = 28)
**flap related**			
necrosis	14.6% (*n* = 2)	18.7% (*n* = 14)	10.8% (*n* = 9)
wound dehiscence	27.2% (*n* = 43)	30.7% (*n* = 23)	24.1% (*n* = 20)
infection	8.2% (*n* = 13)	6.7% (*n* = 5)	9.6% (*n* = 8)
hematoma	1.9% (*n* = 3)	2.7% (*n* = 2)	1.2% (*n* = 1)
seroma	1.3% (*n* = 2)	1.3% (*n* = 1)	1.2% (*n* = 1)
**donor site**			
necrosis	1.3% (*n* = 2)	1.3% (*n* = 1)	1.2% (*n* = 1)
wound dehiscence	27.2% (*n* = 43)	25.3% (*n* = 19)	28.9% (*n* = 24)
infection	0.6% (*n* = 1)	1.3% (*n* = 1)	0.0% (*n* = 0)
granulation problems	7.6% (*n* = 12)	6.7% (*n* = 5)	8.4% (*n* = 7)
hematoma	0.6% (*n* = 1)	0.0% (*n* = 0)	1.2% (*n* = 1)
seroma	4.4% (*n* = 7)	5.3% (*n* = 4)	3.6% (*n* = 3)
**systemic complications**			
pulmonary artery embolism	3.3% (*n* = 5)	2.9% (*n* = 2)	3.7% (*n* = 3)
sepsis	2.0% (*n* = 3)	0.0% (*n* = 0)	3.7% (*n* = 3)

**Table 3 healthcare-11-00403-t003:** Logistic regression analysis showed the significance of the model (chi-square(1) = 7.687, *p* = 0.006 < 0.05, *n* = 158). The variable “season” (*p* = 0.032) was significant. The addition of season in the logistic regression analysis increased the explanatory power of postoperative flap loss from 3.5% to 8.1%, an increase of 4.6%. The R-squared according to Nagelkerke without the addition of the variable season corresponded to 0.089. If the effect size according to Cohen was now calculated using the formula, an effect size of 0.31 was shown, which corresponds to a medium effect. If the season was now added as a variable in the logistic regression, an R-square according to Nagelkerke of 0.205 was shown. The effect strength according to Cohen was 0.51, which corresponded to a strong effect. It is shown that the season with *p* = 0.032 and an Exp(B) = 10.220 had the greatest significance for the development of flap loss, while all other tested variables had no significant effect.

	Regression Coefficient B	Standard Error	Significance	Exp(B)	95% Confidence Interval für Exp(B)	
					lower value	upper value
**Age**	0.022	0.028	0.427	1.023	0.968	1.081
**Gender**	0.597	0.695	0.391	1.816	0.465	7.098
**Body Mass** **Index**	0.008	0.456	0.986	1.008	0.412	2.465
**Diabetes** **Mellitus**	0.348	1.046	0.739	1.416	0.182	10.995
**Hypertension**	0.002	0.829	0.998	1.002	0.198	5.087
**Arteriosclerosis**	−1.074	1.220	0.379	0.342	0.031	3.732
**Anemia**	−0.118	0.776	0.879	0.889	0.194	4.064
**Postoperative antibiotic therapy**	0.613	0.865	0.478	1.846	0.339	10.062
**Smoking**	0.251	0.379	0.508	1.285	0.611	2.701
**Season**	2.267	1.085	**0.037**	**9.655**	1.152	80.918

**Table 4 healthcare-11-00403-t004:** Logistic regression analysis showed the significance of the model (chi-square(1) = 6.636, *p* = 0.010 < 0.05, *n* = 158). The variable temperatures 14 days postoperatively (*p* = 0.023) were significant. By adding the variable 14 days postoperative temperatures into the logistic regression analysis, the explanatory power of flap loss postoperatively grew from 3.5% to 7.5%, an increase of 4.0%. The R-squared according to Nagelkerke without the addition of the variable 14 days postoperative temperatures corresponds to 0.089. If the effect size according to Cohen was now calculated using the formula, an effect size of 0.31 was shown, which corresponded to a medium effect. If the variable 14 days postoperative temperatures was now added to the logistic regression, an R-squared according to Nagelkerke of 0.189 was shown. The effect strength according to Cohen was 0.49, which corresponded to a strong effect. It is shown that temperatures 14 days postoperatively were significant for the development of postoperative flap loss with p = 0.023 (Exp(B) = 1.161).

	Regression Coefficient B	Standard Error	Significance	Exp(B)	95% Confidence Interval for Exp(B)	
					lower value	upper value
**Age**	0.025	0.028	0.378	1.025	0.970	1.083
**Gender**	0.688	0.699	0.325	1.989	0.506	7.823
**Body Mass** **Index**	0.117	0.448	0.793	1.124	0.468	2.703
**Diabetes** **Mellitus**	0.086	1.049	0.935	1.089	0.139	8.515
**Hypertension**	−0.056	0.822	0.945	0.945	0.189	4.735
**Arteriosclerosis**	−1.018	1.210	0.400	0.361	0.034	3.867
**Anemia**	−0.153	0.755	0.840	0.858	0.196	3.769
**Postoperative antibiotic therapy**	0.552	0.830	0.506	1.737	0.341	8.845
**Smoking**	0.425	0.381	0.265	1.529	0.725	3.225
**Average** **temperature 14 days postop.**	0.144	0.065	**0.027**	**1.155**	1.017	1.312

## Data Availability

Not applicable.

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
