# Peer review of "Seasonal Impact on Free Flap Surgery in Terms of Flap Loss and Wound Healing Disorders: A Retrospective Cohort Study of 158 Free Flaps"

_healthcare, 2023, doi:10.3390/healthcare11030403_

Round 1

Reviewer 1 Report

Dear authors,

I read with interest your paper "Seasonal impact on free flap surgery in terms of flap loss and wound healing disorders: a retrospective cohort study of 158 patients over a period of one year".

Despite the study is monocentric and retrospective, I find the topic extremely interesting and well structured. Data are clearly presented.

It would be interesting to compare the results to centers from other countries: summer temperature in Germany are not very hot, probably patients who are used to living in warmer weathers would not get the same risk of dehidratation and trombosis at those temperature.

I reccomend pubblication.

Kind regards,

Rossella Sgarzani

Reviewer 2 Report

This study is aimed to to investigate the seasonal effects on the occurrence of postoperative flap loss and wound healing disorders. Overall it is an interesting paper, but major reporting issues must be addressed.

Please consider to delete "over a period of one year" in the title OR add the year you collected the data.

Introduction fails to support the aim. Authors must organize a sound argument in order to fully support the aim, in fact, they have included interesting info about thrombotic etiology in the Discussion.

Methods are poorly describe, it is necessary to use STROBE statement to describe this section.

Results are redundant, authors must include the results on text OR table, not  both. Moreover, it is not necessary to repeat how you perform the statistical analysis if you described adequately in the Methods.

Discussion provides relevant info on the etiology behind the flap loss, but they only mentioned anecdotal literature. The relationship between thrombosis and season has been studied more extensively and it is interesting explanation to ellaborate in the Discussion https://pubmed.ncbi.nlm.nih.gov/?term=summer%5Btiab%5D+season%5Btiab%5D+thromb*%5Btiab%5D&sort=date

Reviewer 3 Report

Authors made an attempt to explore an interesting topic. However I do have few concerns about the study methodology and results

1. There is disparity in the number of patients in Abstract and main manuscript. Is the total 158 flaps in 151 patients?  if so, did some patients undergo more than 1 flap procedure? 

2. Table 3.2 needs revision as the data in different columns are not aligned properly making it difficult to interpret. 

3. Main concern is the lack of clarity on the details of local room temperature where patients were kept/spending their time, which may have more impact on the patient in the perioperative period rather than outdoor temperature. 

4. Authors have acknowledged their limitation of lack of data on perioperative thromboprophylaxis which again makes it difficult to interpret the study observations. 

Round 2

Reviewer 3 Report

Thank you for your explanations. Couple of points to note. 

Line 61: ‘Free flap arthroplasty’. Not sure authors meant to say all flap procedures were part of arthroplasty. The table suggest otherwise.

 As per Table 3.2

Vascular Complications/Arterial thrombosis are higher in the summer group ( 12% and 10.8%) . However, Flap related necrosis and wound dehiscence are higher in the winter group (18.7% and 30.7%). Is there any explanation for this?  
